# Automated EEG-Based Brainwave Analysis for the Detection of Postoperative Delirium Does Not Result in a Shorter Length of Stay in Geriatric Hip Fracture Patients: A Multicentre Randomized Controlled Trial

**DOI:** 10.3390/jcm13226987

**Published:** 2024-11-20

**Authors:** Emma J. de Fraiture, Henk Jan Schuijt, Maryse Menninga, Iris A. I. Koevoets, Tessa F. M. Verheul, Corine W. van Goor, Thomas M. P. Nijdam, Dieuwke. van Dartel, Johannes H. Hegeman, Detlef van der Velde

**Affiliations:** 1Center for Geriatric Trauma, Department of Surgery, St. Antonius Hospital, 3543 AZ Utrecht, The Netherlands; 2Reggeborgh Research Fellowship Group, ZGT Academy, ZGT Hospital, 7609 PP Almelo, The Netherlands; 3Biomedical Signals and System Group, Faculty of Electrical Engineering, Mathematics and Computer Science, University of Twente, 7522 NB Enschede, The Netherlands; 4Center for Geriatric Trauma, Department of Surgery, ZGT Hospital, 7609 PP Almelo, The Netherlands

**Keywords:** postoperative delirium, geriatric, hip fracture, EEG, DeltaScan, Delirium Observation Screening Scale, trauma, RCT

## Abstract

**Introduction**: Delirium in postoperative geriatric hip fracture patients is a serious and often preventable condition. If detected in time, it can be treated, but a delay in the diagnosis and initiation of treatment impairs outcomes. A novel approach to detect delirium is to use point-of-care electro-encephalogram (EEG) recording with automated analysis. In this study, the authors investigated whether screening for delirium with EEG recording and automated analysis resulted in reduced length of stay after surgery and superior screening performance in comparison to the Delirium Observation Screening Scale (DOS). **Methods**: This randomized control trial was conducted at two geriatric trauma centres in the Netherlands. Patients were eligible for inclusion if they were aged 70 years or above, were admitted to the geriatric trauma unit and undergoing surgery for a hip fracture. Patients were randomized to either the intervention (EEG-based brainwave analysis) or control group (DOSS screening tool). Participants were screened for delirium twice a day during three consecutive days starting at day 0 of the surgery, with the first measurement before the surgery. The primary outcome was length of stay. In addition, the screening performance for both automated EEG-based brainwave analysis and DOS was determined. **Results**: A total of 388 patients were included (189 per arm). There were no differences between groups in terms of median hospital length of stay (DOS 7 days (IQR 5.75–9) vs. EEG-based brainwave analysis 7 days (IQR 5–9); *p* = 0.867). The performance of EEG-based brainwave analysis was considerably lower than that of the DOSS in terms of discrimination between patients with and without postoperative delirium. **Conclusions**: Screening for postoperative delirium in geriatric hip fracture patients using automated EEG-based brainwave analysis did not result in a shorter length of stay. Additionally, the results of this study show no clear advantage in terms of the screening performance of EEG-based brainwave analysis over the current standard of care for geriatric patients with a hip fracture.

## 1. Introduction

Delirium is a serious and often preventable condition. Prevention of delirium consists of non-medical interventions (e.g., orientation measures, monitoring day and night rhythm, adequate pain control, hydration, etc.) [1]. Delirium can be treated if detected in time, but a delay in the diagnosis and initiation of treatment impairs patient outcomes [2,3,4,5,6]. Delirium is very common in hospitalized geriatric hip fracture patients, with a reported incidence in the literature ranging between 13% and 56% [7]. The syndrome is associated with prolonged hospitalization, institutionalization and mortality, as well as increased costs [8,9]. Diagnosis and the measurement of the severity of delirium can be challenging, due to its heterogeneous and multifaceted presentation [2,6,10,11]. A considerable lack of agreement is reported in the literature regarding its classification by experts who independently evaluated exactly the same information [11].

Current screening tools have subjective components, which result in a low sensitivity when used in daily practice. For example, the Confusion Assessment Method for Intensive Care Unit (CAM)-ICU has a sensitivity close to 50%, specificity of about 95%, positive predictive value close to 90% and a negative predictive value of about 70% in routine daily practice for patients in the ICU (intensive care unit) [12]. The Delirium Observation Screening Scale (DOS) is commonly used in non-ICU hospital wards. It has a sensitivity of 62%, a specificity of 98%, a positive predictive value of 96% and a negative predictive value of 82% when used to screen older adults that are admitted to a ward after surgery [11,13].

A novel approach to detect delirium is to use electro-encephalogram (EEG) recordings with electrodes and automated analysis [14,15,16]. Such an approach has the advantage of objective measurements [17]. DeltaScan is a point-of-care medical device that uses EEG recordings, specifically delta wave activity, to detect acute encephalopathy and delirium. Despite insufficient evidence in the literature to form definitive conclusions regarding its diagnostic accuracy, two studies have reported on its capacity to diagnose delirium in both intensive care and non-intensive care settings [18,19].

From a healthcare provider perspective, it would be helpful for clinicians to have a more objective tool to detect delirium, especially if the screening performance is better than the gold standard (DOS) [11]. However, the incremental cost of EEG-based brainwave analysis is an important barrier for implementation. From a healthcare economic perspective, the initial investment and operating costs of EEG-based brainwave analysis could be cost-effective if its use results in earlier detection and treatment of delirium, resulting in a shorter hospital length of stay.

The objective of this study was to thoroughly evaluate EEG-based brainwave analysis by DeltaScan against the gold standard (Delirium Observation Screening Scale) in a geriatric hip fracture population. A randomized controlled trail (RCT) was conducted to create two comparable groups, eliminating differences in the treatment group, and minimize bias. In this study, the authors investigated whether screening for delirium with DeltaScan resulted in reduced length of stay after surgery and superior screening performance in comparison to the DOS.

## 2. Materials and Methods

### 2.1. Study Design

This prospective parallel group randomized control trial was conducted at St. Antonius hospital and Ziekenhuisgroep Twente (ZGT) hospital, both level two geriatric trauma centres in an urban setting in the Netherlands. Both St. Antonius and ZGT have a comanaged orthogeriatric care model. The primary attending physician is always either an orthopedic or trauma surgeon. The study period was August 2021 to August 2024. Patients were eligible for inclusion if they were aged 70 years or above, were admitted to the geriatric trauma unit and underwent surgery for a hip fracture. Exclusion criteria were the following: acute macro brain injury < 6 weeks prior to presentation, admittance with a primary neurological or neurosurgical disease or post-anoxic encephalopathy, patients who could not clinically be assessed for delirium (e.g., due to a language barrier or deafness), patients using lithium or clozapine, patients with an (intra)cranial metal plate or a metal device, or with known pre-existing dementia (dementia is associated with EEG abnormalities, including delta wave abnormalities) [20,21]. The study was reported in line with the CONSORT guidelines for randomized controlled trials [22].

### 2.2. Baseline Characteristics

The following characteristics were collected at baseline: hospital where patient was included (i.e., St. Antonius or ZGT), allocation group (either DeltaScan or DOS), age at presentation (years), sex, fracture type (i.e., femoral neck, pertrochanteric, subtrochanteric or periprosthetic), mechanism of injury (i.e., fall from standing position, fall from height (>0.5 m), pedestrian vs. moving vehicle, traffic accident or other), specialty of attending physician (trauma surgery or orthopedic surgery), prior diagnosis of delirium in medical history (i.e., yes, no, unknown), living situation (i.e., independent, at home with daily ADL care, nursing home, other, unknown).

### 2.3. Study Procedures

After written informed consent was obtained, patients were assigned at random to either the intervention (DeltaScan) or control group (DOS). Patients were enrolled consecutively with an allocation ratio of 1:1. Randomization was carried out using computer generated numbers. A 1:1 randomization was chosen because in general, it offers the most efficiency with the least ethical and study integrity concerns [23]. Participants were screened for delirium twice a day using during three consecutive days.

### 2.4. Intervention Group

Participants in the intervention group were screened for delirium twice a day using the DeltaScan. The first screening took place preoperatively and continued during three consecutive days (Figure 1). The Prolira (Utrecht, The Netherlands) DeltaScan^®^ Brain State Monitor is a CE-marked and EU MDR-approved medical device designed to detect acute encephalopathy, including delirium by analyzing brain activity. It consists of the DeltaScan Patch R1.1, a disposable electrode patch, and the DeltaScan Monitor R2, a portable device containing hardware and software for real-time brain activity recognition [24].

### 2.5. Control Group

Participants in the control group were screened for delirium twice a day using the DOS scale on the same time points as the intervention group (Figure 1).

### 2.6. Outcomes

The primary outcome for this study was hospital length of stay (days), defined as the number of days patients were admitted (i.e., from the moment of admission until discharge from the hospital, or in-hospital death). Secondary outcomes included hospital length of stay until medically ready for discharge (MRD), defined as the number of days patients were admitted, until the attending physician deemed the patient MRD. This approach was adopted because, in the Netherlands, the length of hospital stay can be extended due to the limited availability of skilled nursing facility placements. Measuring the time until MRD may offer insights into this phenomenon. Additionally, time between admission and a diagnosis of delirium (days) and time between the surgery and a diagnosis of delirium (days) was documented for patients who developed delirium.

Delirium was diagnosed by a geriatrician based on clinical presentation and assessment. Patients diagnosed with delirium were treated according to the Dutch standard national delirium guideline and treatment protocol. This guideline states that delirium is diagnosed using the DMS criteria for delirium, sometimes in combination with the Confusion Assessment Method (CAM).

The treatment protocol is based on guidelines of the Society of Critical Care Medicine (SCCM) [25], European Society of Anaesthesiology (ESA) [26] and the American Geriatric Society (AGS) [27].

### 2.7. Data Collection and Data Management

Patient characteristics, measurements and outcomes were documented in case report forms using the Research Electronic Data Capture (REDCap; version 14.0.19; Nashville, TN, USA) application [28]. Data were encoded and stored in a password-protected database at St. Antonius hospital with restricted access to authorized researchers only. Data were entered once. Prior to database locking, the quality of the entered data was evaluated by checking the entries for random patients. In addition, after data collection, outliers were examined for possible erroneous data entries.

For the statistical analysis of continuous data, normality was determined by visual inspection of the boxplots. The threshold for statistical significance in all statistical tests was a two-sided *p*-value < 0.05. For descriptive statistics, a mean and standard deviation were reported for continuous data if they followed a normal distribution. For non-normally distributed continuous data, the median and quartiles were reported. Categorical data were presented as a number with a percentage. For univariable analysis between groups, Student’s *t*-test or the Mann–Whitney U-test was conducted (for normal and non-normally distributed data, respectively). For categorical data, a Chi-squared test or Fisher’s Exact test was conducted, as appropriate.

To adjust for covariates, multiple linear regression was performed for the primary outcome of hospital length of stay. In addition to the primary outcome, a multiple linear regression was performed for the secondary outcome of time between admission and the moment patients were MRD. Based on expert opinion and baseline characteristics, the decision was made to include the following variables in the model: hospital of inclusion, treating physician, age at admission, female sex, prior delirium in medical history and being in DeltaScan group (compared with DOS control group). All variables were entered in the model simultaneously. A complete case analysis was performed (no imputation) in 369 cases. A total of 6 predictors were included in the analysis, resulting in 61.5 cases per variable, which is well above the recommended 30 to 50, following the conventional method for sample size calculation in multiple linear regression [29]. Multivariable analysis of time to diagnosis of delirium was not possible due to a lack of statistical power for this subgroup of 70 patients. Unstandardized coefficients were presented as β, with their corresponding 95% confidence interval.

Additionally, the screening performance was determined by calculating the sensitivity, specificity, false negative rate and false positive rate for the highest score (either DOS or DeltaScan) that was measured in the 3 days after surgery. The area under the curve (AUC) was calculated using ROC analysis. In general, an AUC of 0.5 suggests no discrimination (i.e., ability to diagnose patients with and without delirium based on the test), 0.7 to 0.8 is considered acceptable, 0.8 to 0.9 is considered excellent, and more than 0.9 is considered to be outstanding [30]. All data were analyzed using SPSS, version 29.0 (SPSS, Chicago, IL, USA) [31].

### 2.8. Sample Size Calculation

The sample size calculation is based on the detection of a difference in primary outcome (i.e., hospital length of stay) between the DOS group and DeltaScan group. Detailed calculations for the study sample size are presented in Appendix B. A total sample size of 388 (194 per arm) was required to detect a one-day difference in length of stay between groups.

### 2.9. Ethical Approval

The study was approved by the medical ethical committee MEC-U, Utrecht, the Netherlands, under protocol no. R21.018 and registered by the Central Committee on Research Involving Human Subjects in the Netherlands under protocol no. NL76875.100.21. The study adheres to the ethical principles outlined in the Declaration of Helsinki and was conducted in accordance with ISO14155, ICH GCP guidelines and all local laws and regulations [32,33,34].

## 3. Results

### 3.1. Patient Recruitment

During the inclusion period, 1737 patients were admitted to St. Antonius Hospital and ZGT Hospital and were screened for inclusion (Figure 2). Of these, 408 provided informed consent and were included in the study. The remaining patients either did not meet the inclusion criteria, declined participation or were not asked for informed consent due to the nurse being preoccupied. Of the patients included, 20 were excluded from the final analysis. Reasons for exclusion included nonoperative palliative management, withdrawal of consent, incomplete measurements or death during surgery.

### 3.2. Baseline Characteristics

Baseline characteristics are shown in Table 1. St. Antonius hospital included 326 (84%) patients, whereas ZGT hospital included 62 (16%) patients. The median age in the study population was 81 years (IQR 76–86), with 235 (61%) patients being female. The most common fracture types were femoral neck 227 (59%) and pertrochanteric fractures 146 (38%). The most common trauma mechanisms were falls from a standing position (325) (84%), or a bicycle accident (39) (10%). There were no significant differences between the DeltaScan and DOS group in baseline characteristics, with the exception of the treating physician. Patients in the DeltaScan group were more often treated by an orthopedic surgeon in comparison to the DOS group (16% vs. 30%, respectively; *p* < 0.001).

### 3.3. Patient Outcomes

In the univariable analysis, there were no differences between groups in terms of median hospital length of stay (DOS 7 days (IQR 5.75–9) vs. DeltaScan 7 days (IQR 5–9); *p* = 0.867) (Table 2). For the secondary outcome of hospital length of stay until MRD, there were also no differences between groups (DOS 6 days (4.75–7) vs. DeltaScan 5 days (4–6.5); *p* = 0.089). Additionally, there were no differences in time between admission and diagnosis of delirium or the time between surgery and diagnosis of delirium. A total of six patients (1.5%) died in-hospital, with no significant differences between the DOS or DeltaScan group.

After correcting for covariables in the multiple linear regression analysis for the primary outcome, there was no difference between DeltaScan and DOS in terms of hospital length of stay (Table 3). A higher age at presentation (β = 0.140, *p* < 0.001) was shown to increase the length of hospital stay, as did treatment by an orthopedic surgeon rather than a trauma surgeon (β = −1.215, *p* = 0.030). The other covariates in the analysis (i.e., hospital of presentation, sex and prior delirium in medical history) were not associated with hospital length of stay.

However, for the secondary outcome for hospital length of stay until MRD, there was a significant difference in favour of the DeltaScan group (β = −0.840, *p* = 0.033). A higher age at presentation was again associated with an increase in the length of stay until MRD (β = 0.092, *p* = 0.002). Treatment by an orthopedic surgeon rather than a trauma surgeon (β = −1.356, *p* =0.006) also increased time to MRD. The other covariates in the analysis were again not associated with length of stay.

### 3.4. DeltaScan and DOS Scores in Relation to Postoperative Delirium

A total of 70 patients (18%) were diagnosed with postoperative delirium during their stay. There was no significant difference between the DOS group and DeltaScan group in the incidence of postoperative delirium (DOS; n = 33; 17% vs. DeltaScan n = 37; 19%, *p* = 0.692). The results of the measurements in both groups are visualized in Figure 3a,b, and presented in Appendix A. In the DeltaScan group, patients with a diagnosis of delirium during admission had statistically significant higher DeltaScan scores at all points in time in comparison to patients without delirium. Similarly, the DOS group had statistically significant higher scores at all points in time. For the morning measurement on day 1, there was a large amount of missing data for both the DeltaScan (32%) and DOS group (40%). The reason for this is that some patients were still in the operation theatre or in the recovery room, where the measurement could not be performed.

Both DOS and DeltaScan tests are considered ‘positive’ for delirium if the relevant score ≥ 3 [13,16]. The performance of DOS and DeltaScan are presented in Table 4. In terms of screening performance, for the DOS group, a sensitivity of 0.758, specificity of 0.919, false negative rate of 0.242 and false positive rate of 0.081 was found. The area under the curve was 0.838, indicating excellent discrimination. For the DeltaScan group, a sensitivity of 0.919, specificity of 0.408, false negative rate of 0.081 and false positive rate of 0.592 was found. The area under the curve was 0.663, indicating a poor discrimination. Because of the poor performance of DeltaScan with a cut-off of three, the performance with a cut-off of four and five was also calculated. Using a cut-off of four, a higher discrimination was found (0.754), whereas a cut-off of five resulted in lower discrimination (0.635).

## 4. Discussion

### 4.1. Red Line and Take-Home Message

The aim of this randomized controlled trial was to investigate whether screening for delirium with automated EEG-based brainwave analysis for geriatric patients with hip fractures resulted in reduced length of stay after surgery and superior screening performance in comparison to DOS. The results of this study demonstrated that automated EEG-based brainwave analysis as assessed by DeltaScan screening did not result in a shorter length of stay. Additionally, the performance of DeltaScan was considerably lower than DOS in terms of discrimination between patients with and without postoperative delirium.

### 4.2. Comparison with Previous Literature

Two studies have previously investigated DeltaScan in a cohort study [18,19]. Aben et al. conducted a feasibility study for DeltaScan in comparison to a delirium screening tool (CAM-ICU). The study included 20 intubated ICU patients, 17 of whom developed delirium during admission. There were no statistically significant differences between the DeltaScan and CAM-ICU measurements in diagnosing delirium, although the study was likely underpowered to detect statistically significant differences [19].

Ditzel et al. conducted a large multicenter cohort study that included 223 ICU patients and 267 non-ICU patients across 10 hospitals. The aim was to investigate screening performance in patients 60 years or above for both acute encephalopathy and delirium, using the 10 min delirium interview as a reference. For delirium, sensitivity and specificity were 61% (95% CI = 52%−69%) and 72% (95% CI = 67%−77%). The AUC for delirium was 0.73 (95% CI = 0.65−0.81) for the non-ICU patients [16].

The results found by Ditzel et al. show a slightly higher performance than the results of this RCT, where a sensitivity of 0.919, specificity of 0.408 and AUC of 0.663 was found. The reason for this is likely a difference in setting between the studies. Unfortunately, it is not stated in the paper by Ditzel et al. what the reason for admission to the hospital was, or what department these patients were admitted to, other than non-ICU [16].

### 4.3. Limitations

To our knowledge, this is the first powered multicentre randomized controlled study conducted to evaluate the performance of automated EEG-based brainwave analysis using DeltaScan against a validated delirium screening tool. However, some limitations of this study should be mentioned. First, this study was not powered to detect differences in secondary outcomes such as time to diagnosis of delirium. For this reason, the analysis of secondary outcomes should be interpreted with caution. This will be discussed in the ‘recommendations for future research’ section of this manuscript.

Second, there was a baseline imbalance between the intervention group and control group in terms of attending specialist (Table 1). Given that the allocation of patients to either intervention or control group was achieved electronically and at random, this is most likely coincidental. Although this baseline characteristic was associated with hospital length of stay, it was possible to corrected for this effect in the multivariable analysis.

Third, although this study found a relatively poor overall performance of DeltaScan in comparison to DOS, a significant and clinically relevant difference was found in time to MDR. It should be noted that this is a somewhat subjective outcome measure, prone to performance bias. Therefore, this result should be interpreted with caution.

Last, because of practical reasons this study was not blinded, possibly causing observer bias. Randomization was performed by the research nurse. Because of the nature of the intervention, it was not possible to blind the patient or the nurse. The diagnosis of delirium was given by a geriatrician, adhering to the national guideline for the diagnosis of delirium, irrespective of the score of the screening tool.

### 4.4. Recommendations for Future Research

No difference was found between the groups in terms of time to diagnosis of delirium, though it should be noted that the study was not powered to detect such a difference. DeltaScan may offer a more objective measurement compared to DOS, and it cannot be ruled out that it would indeed result in an earlier diagnosis. It might be tempting to consider this a subject for future studies. However, a powered study to detect a clinically relevant difference in a population with an incidence of delirium of around 18% would require screening a vast number of patients. The required investment in terms of study personnel, time and cost alone would make such a study unfeasible, and it is doubtful whether it would produce clinically relevant results. It seems unlikely that DeltaScan will become standard-of-care for the geriatric trauma population studied here. Even after changing the cut-off for DeltaScan, the screening performance was inferior to DOS (Table 4).

However, this technique may still be useful in settings where patients are at high risk for delirium, but unable to communicate (e.g., intubated patients in the intensive care unit).

### 4.5. Implications for Clinical Practice

The results of this RCT show no clear advantage in terms of screening performance of DeltaScan over the current standard of care for geriatric patients with a hip fracture. Besides performance, several other barriers to implementation should be discussed. One downside to DeltaScan screening is the use of plastic disposables, two or three per patient per day. As stated in the Netherlands Green Deal, healthcare providers should opt for reusable over disposable whenever possible [35]. Physicians and researchers should apply the principles of the circular economy (reduce, reuse and recycle) to healthcare [36]. The use of the DeltaScan disposables is not in line with this policy.

Another barrier to the implementation of DeltaScan is cost. The purchase price of a DeltaScan monitor is EUR 9,999.00. The required disposable patches for the EEG are EUR 19.60 per patient per measurement. For an average geriatric trauma unit treating 400 hip fracture patients a year, performing DeltaScan measurements twice a day for three consecutive days after surgery for every patient would result in an increase in healthcare cost of EUR 47,184.00 every year, in addition to the purchase costs of the required monitor(s). Although it should be noted that this was not a cost-effectiveness study, the primary outcome shows no difference in length of stay. It is reasonable to assume that DeltaScan is more expensive than DOS, considering that the latter is available for free.

Following the results of this RCT, widespread implementation of DeltaScan in the context of geriatric trauma units seems unlikely. However, DeltaScan screening may still prove helpful in other contexts. For example, intubated patients in the ICU are at high risk for delirium, as communication with these patients is challenging.

## 5. Conclusions

In conclusion, the results of this study demonstrated that automated EEG-based brainwave analysis using DeltaScan screening did not result in a shorter length of stay for geriatric patients undergoing surgery for a hip fracture. Additionally, the performance of DeltaScan was considerably lower than the current standard of care (DOS) in terms of discrimination between patients with and without postoperative delirium.

## Figures and Tables

**Figure 1 jcm-13-06987-f001:**
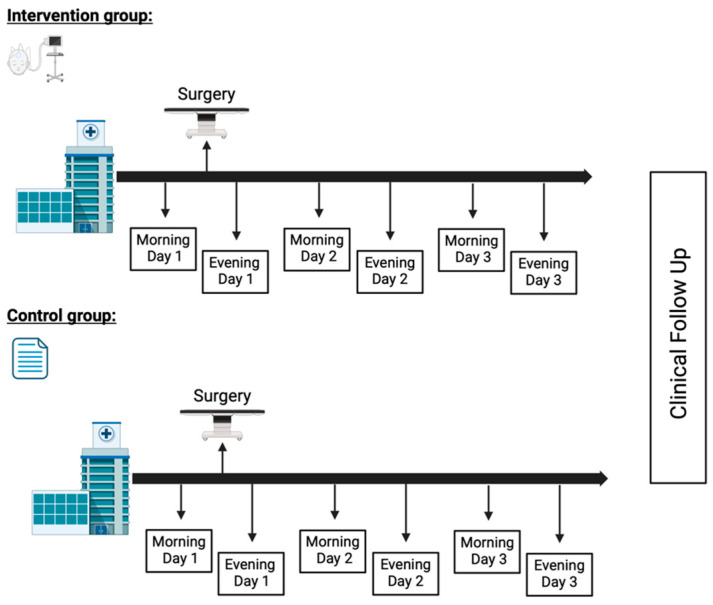
Timeline of study procedures.

**Figure 2 jcm-13-06987-f002:**
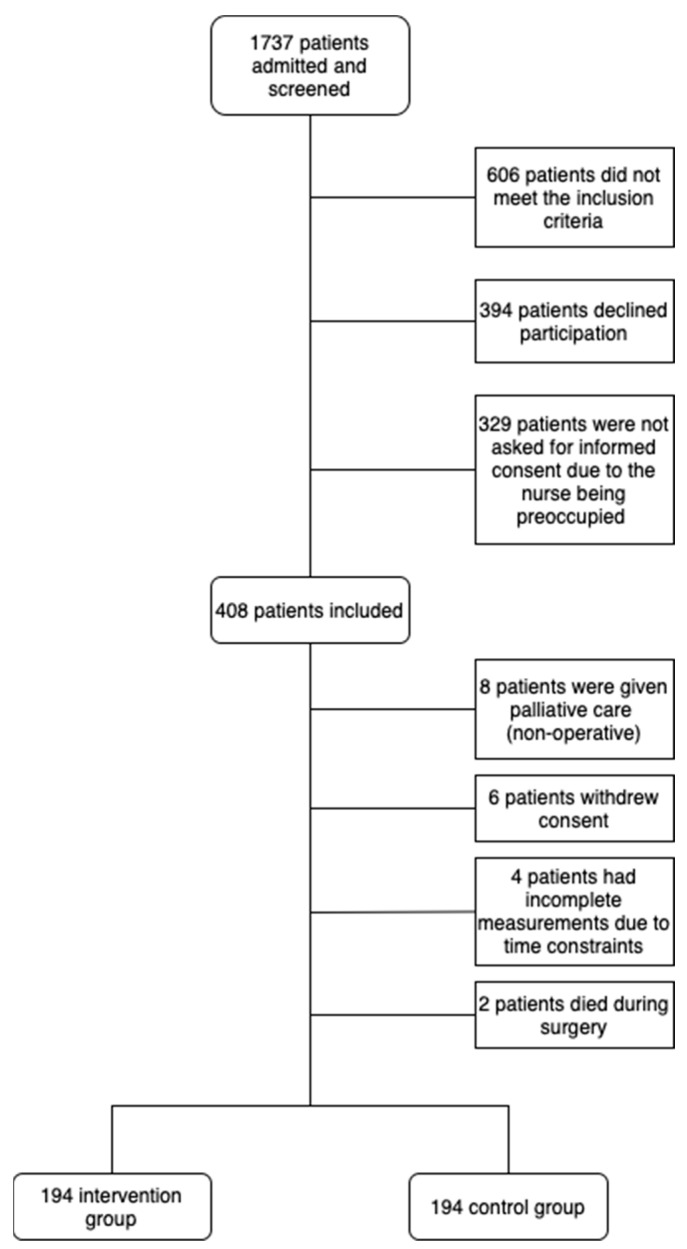
Flowchart of patient inclusion.

**Figure 3 jcm-13-06987-f003:**
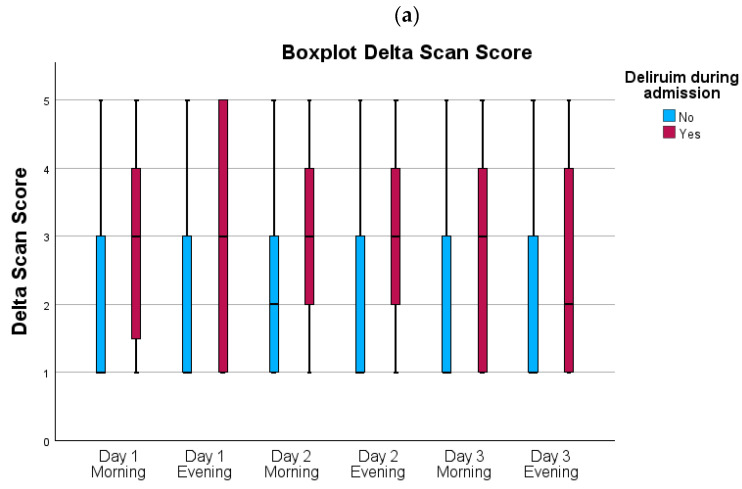
(**a**) Boxplots of DeltaScan scores; (**b**) boxplots of DOS scores. *—outliers.

**Table 1 jcm-13-06987-t001:** Baseline characteristics.

Baseline Characteristic	Total (*n* = 388)	DOS Control (*n* = 194)	DeltaScan (*n* = 194)	*p*-Value
Hospital; *n* (% ^1^)				0.406
St. Antonius	326 (84)	160 (83)	166 (86)	
ZGT	62 (16)	34 (18)	28 (14)	
Age in years; median (IQR)	81 (76–86)	81 (75–86)	81 (76–86)	0.333
Female sex; *n* (%)	235 (61)	110 (57)	125 (64)	0.119
Fracture type; *n* (%)				0.847
Femoral neck	227 (59)	114 (59)	113 (58)	
Pertrochanteric	146 (38)	72 (37)	74 (38)	
Subtrochanteric	6 (2)	4 (2)	2 (1)	
Periprosthetic	9 (2)	4 (2)	5 (3)	
Surgical intervention; *n* (%)				0.876
Hemiarthroplasty	166 (43)	86 (44)	80 (41)	
Intramedullary nailing	140 (36)	71 (37)	69 (36)	
Dynamic hip screw	40 (10)	19 (10)	21 (11)	
Total hip arthroplasty	33 (9)	14 (7)	19 (10)	
Revision arthroplasty	9 (2)	4 (2)	5 (3)	
Mechanism of injury; *n* (%)				0.104
Fall from standing height	325 (84)	170 (88)	155 (80)	
Fall from bicycle or cycling accident	39 (10)	17 (9)	22 (11)	
Fall from height > 0.5 m	18 (5)	5 (3)	13 (7)	
Pedestrian vs. motor vehicle	1 (0)	1 (1)	0 (0)	
Other trauma mechanism	5 (1)	1 (1)	4 (2)	
Treating physician; *n* (%)				<0.001
Trauma surgeon	300 (77)	164 (85)	136 (70)	
Orthopedic surgeon	88 (23)	30 (16)	58 (30)	
Living situation prior to fracture; *n* (%)				0.333
At home, independent	334 (86)	169 (87)	165 (85)	
At home, with ADL assistance	39 (10)	20 (10)	19 (10)	
Institutional care facility	12 (3)	5 (3)	7 (4)	
Other	3 (1)	0 (0)	3 (2)	
Prior diagnosis of delirium; *n* (%)				0.470
Yes	24 (6)	13 (7)	11 (6)	
No	345 (89)	174 (90)	171 (88)	
Unknown	19 (5)	7 (4)	12 (6)	

^1^ All percentages were rounded to closest integer.

**Table 2 jcm-13-06987-t002:** Univariable analysis of patient outcomes.

Patient Outcome	DOS Control (n = 194)	DeltaScan (n = 194)	*p*-Value
Total hospital length of stay (days); median, IQR	7 (5.75–9)	7 (5–9)	0.867
Time between admission and medically ready for discharge (days); median, IQR	6 (4.75–7)	5 (4–6.5)	0.089
Diagnosis of delirium; *n* (%)	33 (17)	37 (19)	0.597
Time between admission and diagnosis of delirium (days); median, IQR	2 (1–3)	2 (1–3)	0.853
Time between OR and diagnosis of delirium (days); median, IQR	1 (0–1)	1 (0–2)	0.540

**Table 3 jcm-13-06987-t003:** Results of multiple linear regression analysis for hospital length of stay.

Hospital Length of Stay			
Variable	β	95% CI of β	*p*-value
Hospital (ZGT compared to SAZ)	0.288	−0.908 to 1.484	0.637
DeltaScan group (compared to DOS)	−0.557	−1.439 to 0.325	0.215
Age at presentation to the ED	0.140	0.074 to 0.207	<0.001
Female sex	−0.389	−1.286 to 0.507	0.394
Treating physician; trauma surgeon * (compared to orthopeadic surgeon)	−1.215	−2.314 to 0.117	0.030
Prior delirium in medical history	0.551	−1.215 to 2.317	0.540
**Hospital length of stay until the moment patients were medically ready for discharge**			
Variable	β	95% CI of β	*p*-value
Hospital (ZGT compared to SAZ)	1.052	0.001 to 2.103	0.050
DeltaScan group (compared to DOS)	−0.840	−1.612 to −0.069	0.033
Age at presentation to the ED	0.092	0.034 to 0.150	0.002
Female sex	−0.467	−1.252 to 0.318	0.243
Treating physician; trauma surgeon * (compared to orthopeadic surgeon)	−1.356	−2.314 to −0.0397	0.006
Prior delirium in medical history	0.166	−1.375 to −1.708	0.832

Abbreviations. ZGT: Ziekenhuisgroep Twente; SAZ: St. Antonius Ziekenhuis; β: unstandardized Beta coëfficiënt; CI: confidence interval; DOS: Delirium Observation Scale; ED: emergency department. * An additional univariable analysis shows that undergoing surgery from trauma surgeons had a median hospital length of stay of 7 days, whereas undergoing surgery from an orthopedic surgeon had a median hospital length of stay of 8 days.

**Table 4 jcm-13-06987-t004:** Screening performance of DOS and DeltaScan.

Test	Sensitivity	Specificity	False Positive Rate	False Negative Rate	Area Under the Curve
DOS Highest score ≥ 3	0.758	0.919	0.081	0.242	0.838
DeltaScan * Highest score ≥ 3	0.919	0.408	0.592	0.081	0.663
DeltaScan ** Highest score ≥ 4	0.865	0.643	0.357	0.135	0.754
DeltaScan ** Highest score 5	0.486	0.783	0.217	0.514	0.635

* DeltaScan is considered positive with a score of 3 or higher [16]. ** Because of the poor performance of DeltaScan with a cut-off of 3, the performance with a cut-off of 4 and 5 was also calculated.

## Data Availability

Data available upon reasonable request.

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
