# Peer review of "Automated EEG-Based Brainwave Analysis for the Detection of Postoperative Delirium Does Not Result in a Shorter Length of Stay in Geriatric Hip Fracture Patients: A Multicentre Randomized Controlled Trial"

_jcm, 2024, doi:10.3390/jcm13226987_

Round 1

Reviewer 1 Report

Comments and Suggestions for Authors

why was DOS selected as the gold standard for comparison, instead of CAM?

what were the limitations of current EEG based tools like DeltaScan?

is there preliminary data /studies that support your assumption that EEG based brainwave could be cost effective?

how do authors plan to account for the lack of consensus (61% agreement) in your study methodology?

what specific advantages do authors anticipate DeltaScan over DOS?

fix the inconsistent use of the term "delirium" by standardizing its capitalization throughout the manuscript

why did the authors focus on geriatric hip fracture population?

include a more clear reason that strengthen the justification for the randomized controlled trial

"golden standard" should be  "gold standard"

excluding patients with preexisting dementia is not explained, even though these patients are at high risk for delirium

provide a reason for excluding patients with dementia

the authors define the primary outcome as "hospital length of stay (in days)" but lacks an explanation on how discharge is determined

address whether the study was blinded and, if not, discuss the potential biases

there is a lack of reason for 1:1 randomization

the method for checking data quality is vague ("checking the entries for random patients")

Author Response

Reviewer 1

Dear reviewer,

Thank you for reviewing our manuscript and providing feedback for improvement. We will address the feedback in a point-by-point manner.

why was DOS selected as the gold standard for comparison, instead of CAM?

Answer: the DOS is a screening instrument used by nurses, whereas the CAM supports diagnosis by a geriatrician. DeltaScan is a screening tool, so it has to be compared to another screening tool. This is specified more clearly in the method section

Changes made:

L156

Delirium was diagnosed by a geriatrician based on clinical presentation and assessment. Patients diagnosed with delirium were treated according to the Dutch standard national delirium guideline and treatment protocol. This guideline states that delirium is diagnoses using the DMS criteria for delirium, sometimes in combination with the Confusion Assessment Method (CAM).

what were the limitations of current EEG based tools like DeltaScan?

Answer:

  1. Limitations include; lack of scientific evidence for effectiveness (discussed in the introduction and discussion)
  2. Cost (discussed in the discussion sections under “implications for clinical practice”)
  3. The use of disposables (discussed in the discussion sections under “implications for clinical practice”)

is there preliminary data /studies that support your assumption that EEG based brainwave could be cost effective?

Answer:

There is no preliminary data to support this assumption. This product (DeltaScan) was marketed by the producer Prolira with this assumption in mind. Because there was no evidence to support this, and healthcare providers wanted to know if it was true, this RCT was designed.

how do authors plan to account for the lack of consensus (61% agreement) in your study methodology?

Answer:

This is a valid point. The agreement of 61% is incorrect, and changes have been made accordingly. The study by Numan et al (doi: 10.1111/jgs.14933) shows a 79% agreement (kappa of 0.61). Unfortunately, this uncertainty is inherent to delirium diagnosis, which illustrates the need for more robust diagnostic tools.

Changes made:

L60 Considerable lack of agreement is reported in literature regarding its classification by experts who independently evaluated exactly the same information.

what specific advantages do authors anticipate DeltaScan over DOS?

Before this study; the idea was that advantages of DeltaScan would include: a more objective & specific tool to detect delirium, possibly easier to use by nurses, and its ability to screen for delirium in patients that are unable to communicate.

This is discussed in L73-87 of the introduction.

fix the inconsistent use of the term "delirium" by standardizing its capitalization throughout the manuscript

The term “delirium” is not capitalized in the manuscript, with the exception of the term “Delirium Observation Scale” (DOS), because it is a name and hence should be capitalized.

why did the authors focus on geriatric hip fracture population?

To assess the clinical advantages of screening for delirium with DeltaScan, we aimed to include a homogeneous population of surgically treated geriatric trauma patients, because they are frail and have a very high risk of postoperative delirium. If our study would have concluded that the DeltaScan is superior in screening for delirium and therefore causes a shorter stay in the hospital, the study could be repeated for different populations.

include a more clear reason that strengthen the justification for the randomized controlled trial

Changes made

L90 The objective of this study was to thoroughly evaluate EEG-based brainwave analysis by DeltaScan against the golden standard (Delirium Observation Screening Scale) in a geriatric hip fracture population. A randomized controlled trail (RCT) was conducted to create two comparable groups, eliminating differences between treatment group and minimize bias.

"golden standard" should be  "gold standard"

Thank you for this suggestion, changes were made accordingly.

excluding patients with preexisting dementia is not explained, even though these patients are at high risk for delirium. provide a reason for excluding patients with dementia

Patients with dementia often have abnormal Delta wave activity on EEG. Because DeltaScan uses Delta wave activity, it is unsuitable to differentiate between dementia and delirium in this phase of the research.

Changes made

“[…] , or with known pre-existing dementia (dementia is associated with EEG abnormalities, including delta wave abnormalities).”

the authors define the primary outcome as "hospital length of stay (in days)" but lacks an explanation on how discharge is determined.

Discharge was defined as the moment as the number of days patients were admitted, (i.e. from the moment of admission until discharge from the hospital, or in-hospital death).

Changes made

L147 The primary outcome for this study was hospital length of stay (days), defined as the number of days patients were admitted, (i.e. from the moment of admission until discharge from the hospital, or in-hospital death).

address whether the study was blinded and, if not, discuss the potential biases

We have addressed this issue in the limitations section of the discussion.

Changes made:

Last, because of practical reasons this study was not blinded, possibly causing observer bias. Randomization was performed by the research nurse. Because of the nature of the intervention, it was not possible to blind the patient or the nurse. The diagnosis of delirium was done by a geriatrician, adhering to the national guideline for the diagnosis of delirium, irrespective of the score of the screening tool.

there is a lack of reason for 1:1 randomization

changes made

 L30 In general, a 1:1 randomization offers the most efficiency with the least ethical and study integrity concerns.

the method for checking data quality is vague ("checking the entries for random patients")

We elaborated on this a little bit more

Changes made L 171 Prior to database locking, the quality of the entered data was evaluated by checking the entries for random patients. In addition, after data collection, outliers were examined for possible erroneous data entries.

Reviewer 2 Report

Comments and Suggestions for Authors

This article reports on a multicenter randomized controlled trial evaluating the effectiveness of automated EEG-based brainwave analysis (DeltaScan) for detecting postoperative delirium in geriatric hip fracture patients. The study found no significant difference in hospital length of stay between DeltaScan and the traditional DOS, with DeltaScan showing lower screening performance.
the study is well written and nice to read

i have some comments:

Abstract : replace discussion to conclusion 

introduction
i suggest to remove subheadings in the manuscript and present the introduction in multiple paragraphs 
put reference citation between brackets over all the manuscript
give incidence of delirium in hip fracture and risk factors, prevention and how can be treated , and the role of early detection and why it's important to use these screening tool etc briefly in short paragraph
will these detection tools can lead to change or modification of management ?  all these will be nice to add to re-enforce the introduction

methodolgy and results: will presented but can authors provide mean time from fracture (or admission) to surgery as it can interesting to know as it may affect the results.

can authors also provide data for place of discharge in percentage with p value for each group >> home or nursing home , rehab center  
as type of discharge can also affect results , sometimes rehab center need more organization and more hospital stay until place is available in the center . From our experience in France, Patients with hip fracture, can be wait for discharge untill place is available which can affect results while patients who discharged at home can be discharged earlier.

discussion : very nice discussion specially cost issues
but i recommend to adress the limitation of the study

Author Response

reviewer 2

Dear reviewer,

Thank you for reviewing our manuscript and providing feedback for improvement. We will address the feedback in a point-by-point manner.

This article reports on a multicenter randomized controlled trial evaluating the effectiveness of automated EEG-based brainwave analysis (DeltaScan) for detecting postoperative delirium in geriatric hip fracture patients. The study found no significant difference in hospital length of stay between DeltaScan and the traditional DOS, with DeltaScan showing lower screening performance.
the study is well written and nice to read

i have some comments:

Abstract : replace discussion to conclusion 

Changes made accordingly

introduction
i suggest to remove subheadings in the manuscript and present the introduction in multiple paragraphs 

Changes made accordingly

put reference citation between brackets over all the manuscript

Changes made accordingly

give incidence of delirium in hip fracture and risk factors, prevention and how can be treated , and the role of early detection and why it's important to use these screening tool etc briefly in short paragraph
will these detection tools can lead to change or modification of management ?  all these will be nice to add to re-enforce the introduction

Changes made

L50-61

Delirium is a serious and often preventable condition. Prevention of delirium consists of non-medical interventions (e.g. orientation measures, monitoring day and night rithm, adequate pain control and hydration, etc.). Delirium  can be treated if detected in time, but a delay in diagnosis and initiation of treatment impairs patients outcomes4–8A delirium is very common in hospitalized geriatric hip fracture patients, with reported incidence in literature ranging between 13% and 56%.9 The syndrome is associated with prolonged hospitalization, institutionalization and mortality, as well as increased costs

methodolgy and results: will presented but can authors provide mean time from fracture (or admission) to surgery as it can interesting to know as it may affect the results.

Unfortunately, we do not have this data available.

can authors also provide data for place of discharge in percentage with p value for each group >> home or nursing home , rehab center  as type of discharge can also affect results , sometimes rehab center need more organization and more hospital stay until place is available in the center . From our experience in France, Patients with hip fracture, can be wait for discharge untill place is available which can affect results while patients who discharged at home can be discharged earlier.

In The Netherlands, we have a similar problem with discharge to skilled nursing facilities. To correct for this possible bias, we also included “medically ready for discharge” (MDR) as a secondary outcome measure. This is discussed in L148-150.

discussion : very nice discussion specially cost issues
but i recommend to adress the limitation of the study

We have addressed this issue in the limitations section of the discussion.

Changes made:

Last, because of practical reasons this study was not blinded, possibly causing observer bias. Randomization was performed by the research nurse. Because of the nature of the intervention, it was not possible to blind the patient or the nurse. The diagnosis of delirium was done by a geriatrician, adhering to the national guideline for the diagnosis of delirium, irrespective of the score of the screening tool.

Reviewer 3 Report

Comments and Suggestions for Authors

Thank you for this interesting manuscript. In this study, the use of automated EEG-based brainwave analysis (DeltaScan) for the detection of postoperative delirium in geriatric hip fracture patients did not result in a shorter hospital stay compared to the Delirium Observation Screening Scale (DOS). Moreover, DeltaScan's diagnostic performance in this trial was inferior to that of DOS in terms of specificity and AUC, raising questions about its clinical utility in the geriatric population.

Study Design and Power Analysis
Flaw: The sample size calculation is well documented, but the study lacks power for secondary outcomes, such as the time to delirium diagnosis and other subgroup analyses. The underpowered results should not be overstated in the conclusions section.

Missing Data
There were significant missing data for both DeltaScan and DOS measurements, especially on day 1 (32% and 40% missing data, respectively). This could introduce bias. A more in-depth explanation and justification for handling missing data is required. Did the authors use imputation methods or sensitivity analysis? The lack of data from key time points weakened the internal validity of the study.

Study Outcomes
While the primary outcome (length of hospital stay) was clearly defined, the outcome "medically ready for discharge" (MRD) is subjective and prone to bias. The MRD results suggest bias or an unblinded assessment.

Screening Performance
DeltaScan showed a poor AUC of 0.663 compared with DOS (0.838). The authors explored different cut-off points for DeltaScan, but even the highest-performing cut-off (AUC = 0.754) did not achieve the same level as the DOS. The discussion could place more emphasis on the potential clinical implications of these findings, particularly questioning the clinical utility of DeltaScan, given its high false-positive rate. Should the DeltaScan cut-off be adjusted or is the tool fundamentally limited in the postoperative setting?

Comparison with Existing Literature
The comparison with previous studies is somewhat superficial. The authors referenced two earlier studies using DeltaScan, but did not sufficiently explain how their RCT adds to or contradicts these findings. Moreover, the poor sensitivity and specificity of DeltaScan contrasts with other reports, but this has not been well explored. The authors should engage more deeply with recent literature to highlight how their findings align or diverge from those of other studies on EEG-based delirium detection. A more thorough discussion is needed on why DeltaScan may perform differently in this geriatric postoperative population compared with ICU patients in previous studies.

Conclusions
However, the conclusions do not fully address the clinical significance of these findings. The poor screening performance of DeltaScan is a critical result; however, the authors still suggest that it may have some utility, especially in the ICU setting. This seems to contradict the findings of the present study. The conclusion should be revised to provide a more balanced view, emphasising that DeltaScan does not outperform DOS and is unlikely to be adopted in postoperative settings unless significant improvements are made. Overly optimistic conclusions should be drawn, particularly when the results are not statistically or clinically meaningful.

General
This manuscript has valuable insights, but requires significant improvements in the interpretation of results, handling of bias and missing data, and positioning within the broader literature. Additional clarity on randomisation and a more robust discussion are needed; therefore, major revisions are needed before the manuscript is ready for publication.

sincerely

Author Response

reviewer 3

Dear reviewer,

Thank you for reviewing our manuscript and providing feedback for improvement. We will address the feedback in a point-by-point manner.

Thank you for this interesting manuscript. In this study, the use of automated EEG-based brainwave analysis (DeltaScan) for the detection of postoperative delirium in geriatric hip fracture patients did not result in a shorter hospital stay compared to the Delirium Observation Screening Scale (DOS). Moreover, DeltaScan's diagnostic performance in this trial was inferior to that of DOS in terms of specificity and AUC, raising questions about its clinical utility in the geriatric population.

Study Design and Power Analysis
Flaw: The sample size calculation is well documented, but the study lacks power for secondary outcomes, such as the time to delirium diagnosis and other subgroup analyses. The underpowered results should not be overstated in the conclusions section.

Changes made:

this study was not powered to detect differences in secondary outcomes such as  time to diagnosis of delirium. For this reason, the analysis of secondary outcomes should be interpreted with caution. This will be discussed in the ‘recommendations for future research’ section of this manuscript.

Missing Data
There were significant missing data for both DeltaScan and DOS measurements, especially on day 1 (32% and 40% missing data, respectively). This could introduce bias. A more in-depth explanation and justification for handling missing data is required. Did the authors use imputation methods or sensitivity analysis? The lack of data from key time points weakened the internal validity of the study.

Missing data is discussed in L271-275

For the morning measurement on day 1, there was a large amount of missing data for both the DeltaScan (32%) and DOS group (40%). The reason for this is that some patients were still in the operation theatre or in the recovery room, where the measurement could not be performed.

Imputation for this secondary outcome was not performed as no multivariable analysis was done for this parameter, and data were likely not missing at random.

Study Outcomes
While the primary outcome (length of hospital stay) was clearly defined, the outcome "medically ready for discharge" (MRD) is subjective and prone to bias. The MRD results suggest bias or an unblinded assessment.

This is a valid point. The subjective nature of MRD is discussed in the paper

Third, although this study found a relatively poor overall performance of the DeltaScan in comparison to the DOS, a significant and clinically relevant difference was found in time to MDR. It should be noted that this is a somewhat subjective outcome measure, prone to performance bias. Therefore, this result should be interpreted with caution.

Changes made

Last, because of practical reasons this study was not blinded, possibly causing observer bias. Randomization was performed by the research nurse. Because of the nature of the intervention, it was not possible to blind the patient or the nurse. The diagnosis of delirium was done by a geriatrician, adhering to the national guideline for the diagnosis of delirium, irrespective of the score of the screening tool.

Screening Performance
DeltaScan showed a poor AUC of 0.663 compared with DOS (0.838). The authors explored different cut-off points for DeltaScan, but even the highest-performing cut-off (AUC = 0.754) did not achieve the same level as the DOS. The discussion could place more emphasis on the potential clinical implications of these findings, particularly questioning the clinical utility of DeltaScan, given its high false-positive rate. Should the DeltaScan cut-off be adjusted or is the tool fundamentally limited in the postoperative setting?

The cut-off adjuments are discussed and reported in the results section L282 and Table 4

Because of the poor performance of DeltaScan with a cut-off of 3, the performance with a cut-off of 4 and 5 was also calculated. Using a cut-off of 4, a higher discrimination was found (0.754), whereas a cut-off of 5 resulted in lower discrimination (0.635).

Changes made in limitations section

“Even after changing the cut-off for DeltaScan, the screening performance was inferior to DOS (Table 4).”

Comparison with Existing Literature
The comparison with previous studies is somewhat superficial. The authors referenced two earlier studies using DeltaScan, but did not sufficiently explain how their RCT adds to or contradicts these findings. Moreover, the poor sensitivity and specificity of DeltaScan contrasts with other reports, but this has not been well explored. The authors should engage more deeply with recent literature to highlight how their findings align or diverge from those of other studies on EEG-based delirium detection. A more thorough discussion is needed on why DeltaScan may perform differently in this geriatric postoperative population compared with ICU patients in previous studies.

Though it would be interesting to present a more extensive comparison with existing literature, there are simply not many papers that have previously investigated DeltaScan. All relevant literature regarding performance of DeltaScan has been cited, to the knowledge of the authors. If there are any specific references missing, please let us know which ones they are, we would be very happy to include them in our manuscript.

Conclusions
However, the conclusions do not fully address the clinical significance of these findings. The poor screening performance of DeltaScan is a critical result; however, the authors still suggest that it may have some utility, especially in the ICU setting. This seems to contradict the findings of the present study. The conclusion should be revised to provide a more balanced view, emphasising that DeltaScan does not outperform DOS and is unlikely to be adopted in postoperative settings unless significant improvements are made. Overly optimistic conclusions should be drawn, particularly when the results are not statistically or clinically meaningful.

In the conclusion, the critical result of the poor screening performance is discussed:

In conclusion, the results of this study demonstrated that automated EEG-based brainwave analysis using DeltaScan screening did not result in a shorter length of stay for geriatric patients undergoing surgery for a hip fracture. Additionally, the performance of DeltaScan was considerably lower than the current standard of care (DOS) in terms of discrimination between patients with and without postoperative delirium.”

We agree that overly optimistic conclusions or recommendations should not be included in the paper. Removed:  “It should also be noted that this trail studied the geriatric hip fracture population as a whole. Future research might focus more on the most frail patients.”

General
This manuscript has valuable insights, but requires significant improvements in the interpretation of results, handling of bias and missing data, and positioning within the broader literature. Additional clarity on randomisation and a more robust discussion are needed; therefore, major revisions are needed before the manuscript is ready for publication.

sincerely

Round 2

Reviewer 1 Report

Comments and Suggestions for Authors

The authors answered my questions and recommendations. The work can be published now.

Author Response

Dear colleague,

Thank your for your review of this manuscript.

Henk Jan Schuijt

Reviewer 3 Report

Comments and Suggestions for Authors

the authors made significant improvements and have addressed all my concerns

Author Response

(The authors gave the same response as above.)
